# Control of an Outer Rotor Doubly Salient Permanent Magnet Generator for Fixed Pitch kW Range Wind Turbine Using Overspeed Flux Weakening Operations

Aziz Remli [1], Cherif Guerroudj [2,*], Jean-Frédéric Charpentier [3], Tarek Kendjouh [2], Nicolas Bracikowski [4] and Yannis L. Karnavas [5]

1   Laboratoire de Maitrise des Energies Renouvelables (LMER), Faculté de Technologie, Université de Bejaia, Bejaia 06000, Algerie
2   Laboratoire des Systèmes Electriques Industriels (LSEI), BP No. 32 El-Alia, Bab Ezzouar 16111, Algerie
3   Institut de Recherche de l'Ecole Navale (EA 3634, IRENav), French Naval Academy, 29240 Brest, France; jean-frederic.charpentier@ecole-navale.fr
4   Institut de Recherche en Énergie Électrique de Nantes-Atlantique (IREENA), Université de Nantes, 44602 Saint-Nazaire, France
5   Electrical Machines Laboratory, Department of Electrical and Computer Engineering, Democritus University of Thrace, 671 00 Xanthi, Greece
*   Correspondence: cherif.guerroudj@yahoo.fr

**Abstract:** This paper deals with the analysis of the dynamic performance of a generator with a doubly salient exterior rotor excited by permanent magnets inserted in the stator yoke. The electrical generator works at low speed and is devoted to a wind energy conversion system. Indeed, the studied generator is a robust high-torque machine and can be directly coupled to the turbine blades. It must therefore assure the energy conversion for wind speeds lower or higher than its base speed. In fact, the control technique used in this work covers the two main operating zones: below the base speed and above it. In the first case, the maximum torque per ampere control law is developed; however, when the base speed is reached, the flux decay control law is implemented and, consequently, the system works above the nominal conditions. Fuzzy logic controllers are employed to regulate direct and quadrature machine currents and DC voltage in order to obtain satisfactory regulation performances. The ensemble of the wind turbine and electrical machine with technical control is performed in Matlab/Simulink software. The simulation results obtained show the capability of the machine to operate at variable speeds, ensuring efficient energy conversion under and over the nominal speed.

**Keywords:** wind energy conversion system; outer rotor; DSPMG; MTPA; flux weakening; fuzzy logic controller; direct–drive system

## 1. Introduction

Presently, to face of the depletion of primary energy resources (oil, natural gas, coal, etc.) and for the sake of preserving the environment, many countries are reviewing their energy policies and are increasingly interested in renewable energies, in particular wind energy.

To exploit this potential, various programs for the development of wind energy production have been launched. One of the main objectives of these programs is to reduce the price of the kWh produced but also the price of the investment [1]. In fact, wind energy is only viable if its cost price is competitive. This drop in the price of the kWh produced will have to go through the improvement of the entire conversion chain. To this end, various studies have been undertaken concerning the mechanical part (blades made of composite materials, lightning of the masts, etc.), the automatic part, and diagnostics (prevention of breakdowns, mechanical regulation, etc.). In regard to the electrical part, it is articulated around two major essential axes [2]:

— Energy conversion optimization with variable speed operations;
— Direct coupling of turbine and generator by eliminating the gearbox.

Several wind energy conversion system (WECS) structures have been developed with various control techniques and are mainly: doubly feed induction generators (DFIG) and permanent magnet synchronous generators (PMSG) [3,4]. In the case of large–scale WECS, both DFIG and PMSG are employed and often equipped with a gearbox. Furthermore, power limitation at overrated wind speed is mainly insured by means of pitch control [5,6]. However, the gearbox and the pitch mechanism greatly affect the cost and robustness of the WECS and increase the maintenance requirement. In the case of small–scale WECS, a direct–drive variable–speed PMSG is widely used, and an objective can be to eliminate pitch control system to reduce the system complexity [7,8]. Moreover, if the WECS is equipped with full bridge back to back converters, various control techniques can be employed in order to enhance system flexibility and efficiency. Since the pitch control is eliminated, in small–scale WECS, generator operation at wind overspeed region is only obtained by flux weakening. In fact, this strategy is widely employed for motor applications [9–13], but a gain interest is touched for this application in wind and marine current energy conversion systems [7,14,15].

There are also other special structures, proposed in the literature for small–scale WECS with a power range of 0.5–50 kW and low speed in the range of 30 to 150 rpm, which are classified with respect to their operation as synchronous machines; these machines are called doubly salient permanent magnet (DSPM) machines [16–20]. Indeed, these topologies are not new, but reemerged thanks to progress achieved in electronic and power electronic devices and magnet materials. Designated with the stator and the rotor pole numbers such as DSPM 6/4, 8/6, 12/10, and 36/24, these topologies are not convenient for low–speed applications owing to the dependence between the machine volume and the sizing torque which becomes a serious challenge.

In this context, the outer–rotor doubly salient permanent magnets generator (OR–DSPMG) is proposed for application in the small–scale kW range low speed and direct–drive WECS. Such a structure has been dimensioned in [21,22], for low power WT applications with a low nominal speed of 50 rpm, in order to keep the equipment costs of all the components of the machine at a low level and to be able to build the whole installation more compact and lighter. From a mechanical point of view, an outer–rotor with a low weight and inertia constitute an evident advantage because the turbine blades can be directly mounted on the machine rotor surface. The used OR−DSPMG, presented in Figure 1, is based on the variable reluctance machine in which both the stator and the rotor are toothed to have double saliency. The stator carries small teeth ($N_s$). Stator poles ($N_{ps}$) also carry small teeth in the same way. This solution leads to operating the machine at low speed and producing high torque. The coils of the three phases are wound around the stator poles (concentrated winding) with each phase consisting of four coils connected in series. Rare Earth PMs (Nd-Fe-Br) are placed in the stator yoke and provide the excitation of the machine. In this structure, the rotor is placed outside and the stator inside the machine (external rotor machine). The magnetic circuit of both stator and rotor is formed by M400–50A steel type. Studied machine geometrical parameters (axial length, number of stator pole, number of stator and rotor teeth, stator and rotor yoke thickness and teeth depth, outer rotor radius, slot radius, air gap thickness, and magnet thickness) and PMs magnetization and relative permeability are given in Table A1 (Appendix A).

This article is the continuity of the work done in [22], which consists of dynamic modeling and analysis of dynamic behavior and performance of the OR−DSPMG applied to a wind turbine chain at variable speeds.

The energy conversion system (10 kW−50 rpm direct–drive and grid–connected WECS) is equipped with a low–speed high torque unconventional machine (OR−DSPMG), horizontal axis wind turbine with three glass fiber blades and full bridge back to back pulse width modulation (PWM) rectifier and inverter. The system is designed to operate efficiently for all speed range, i.e., underrated and overrated wind speed, without pitch

control. So, an optimal flux weakening strategy is developed to control the machine and turbine in turbine overspeed operation with constant power under given current and voltage constraint, as achieved in [15] for a marine current energy conversion system based on PMSG. Below rated speed, maximum power point tracking (MPPT) control is released by using perturb and observe algorithm, without need to wind sensor. On the other hand, maximum torque per ampere (MTPA) control is used as a criterion for OR−DSPMG control. This strategy has a proven effectiveness for PMSG through simulation results [7,23,24] and experimental validation [7,24]. The MTPA control is achieved by means of current regulation in d–q reference frame by moving operation point along the MTPA trajectory. Beyond the rated speed, optimal flux weakening control leads to generator operation with the maximum torque under relative current and voltage limitations. These limitations are imposed by the machine and converter and allow power limitation as described in [14]. The corresponding d–q current references are generated and regulated. Indeed, for both MTPA and flux weakening control strategies, fuzzy logic controllers (FLC) are performed for d–q currents regulation due to these demonstrated performances (time response, steeling time, overshoot, and robustness) compared with conventional PI regulators, especially for non-linear systems [25,26].

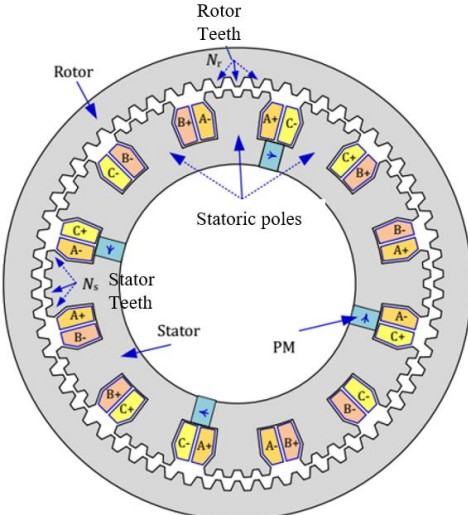

**Figure 1.** Outer rotor toothed doubly salient permanent magnet generator (OR−DSPMG) [22].

The main contribution of this work is the development of MTPA and flux weakening control strategies for low–speed direct–drive WECS equipped with a new machine topology (OR−DSPMG) that can operate at underrated and overrated wind speed.

The paper is divided into five sections; the second section presents the main wind turbine and OR−DSPMG equations. Section 3 shows the implemented control strategies: MPPT−MTPA in partial load region and flux weakening in full load region, respectively. Simulation results are presented and discussed in Section 4; in Section 5, conclusions are presented.

## 2. Wind Turbine and OR−DSPMG Modelling

### 2.1. Wind Turbine Modelling

Aerodynamic power extracted form wind kinetic energy is expressed as follows [27]:

$$P_t = \frac{1}{2}\rho\pi R^2 C_p(\lambda) v_t^3 \tag{1}$$

The power coefficient depends on the tip speed ratio as shown in Figure 2a; tip speed ratio is given by:

$$\lambda = \frac{\Omega R}{v_t} \tag{2}$$

where ρ, R, $C_p$, $v_t$ and Ω are, respectively, the air density, blade turbine radius, power coefficient, wind speed, and mechanical shaft speed (Table A1, Appendix A).

The perturb and observe maximum power point tracking algorithm (PO–MPPT) provide the desired mechanical speed corresponding to the maximum extracted power from wind kinetic energy [28], which is given by:

$$P_{t-max} = \frac{1}{2}\rho \, C_{pmax} \frac{\pi R^5}{\lambda_{opt}^3} \Omega_{MPPT}^3 \qquad (3)$$

When the rated point is reached (Figure 2b), the extracted power is limited in order to avoid the electrical conversion system oversizing. Since the wind turbine is fixed pitch blades, power limitation in the overspeed region is achieved through generator torque control under flux weakening operation.

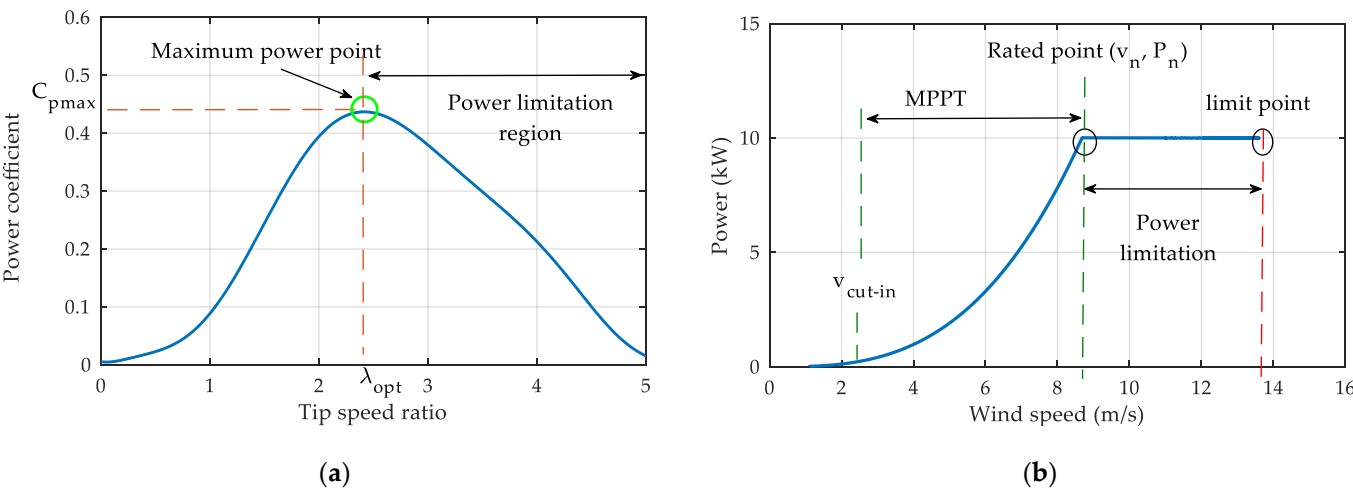

(**a**)  (**b**)

**Figure 2.** Wind turbine characteristics: (**a**) Power coefficient; (**b**) Extracted power vs. wind speed.

### 2.2. OR−DSPMG Modelling

Park transformation will be applied for machine equations in stator reference frame. The Park matrix is given by:

$$P(\theta_e) = \sqrt{\frac{2}{3}} \begin{bmatrix} \cos(\theta_e) & -\sin(\theta_e) \\ \cos\left(\theta_e - \frac{2\pi}{3}\right) & -\sin\left(\theta_e - \frac{2\pi}{3}\right) \\ \cos\left(\theta_e - \frac{4\pi}{3}\right) & -\sin\left(\theta_e - \frac{4\pi}{3}\right) \end{bmatrix} \qquad (4)$$

So, the main equations of the OR−DSPMG in the d−q reference frame are presented in this section; further details about DSPMG modeling can be found in [29,30].

Voltage expressions are given by:

$$\begin{cases} v_d = -\left(R_s + 2w_e M_{dq}\right)i_d + \frac{w_e}{2}(3L_d - L_q)i_q - L_d \frac{di_d}{dt} - M_{dq}\frac{di_q}{dt} \\ v_q = -\left(R_s - 2w_e M_{dq}\right)i_q + \frac{w_e}{2}(3L_q - L_d)i_d - L_q \frac{di_q}{dt} - M_{dq}\frac{di_d}{dt} - \sqrt{\frac{3}{2}}\varphi_1 w_e \end{cases} \qquad (5)$$

with

$$\begin{cases} L_{d,q} = L_0 - M_0 \pm \frac{1}{2}(L_1 + 2M_1)\cos(3\theta_e) \\ M_{d,q} = -\frac{1}{2}(L_1 + 2M_1)\sin(3\theta_e) \\ \theta_e = \int w_e \, dt \end{cases} \qquad (6)$$

Magnetic flux equations are:

$$
\begin{cases}
\varphi_d = L_d i_d + M_{dq} i_q + \sqrt{\dfrac{3}{2}} \varphi_1 \\
\varphi_q = L_q i_q + M_{dq} i_d
\end{cases}
\tag{7}
$$

Electromagnetic torque is written in Equation (8):

$$
T_{em} = -\sqrt{\dfrac{3}{2}}\, N_r \varphi_1 i_q + \dfrac{1}{2} N_r (L_d - L_q) i_d i_q - \dfrac{1}{2} N_r M_{dq} \left( i_d^2 - i_q^2 \right)
\tag{8}
$$

Mechanical equation is expressed as:

$$
J_t \dfrac{d}{dt} \Omega = T_{em} - T_m - f_v \Omega.
\tag{9}
$$

Active and reactive machine powers are expressed by the following equations:

$$
\begin{cases}
P_a = v_d i_d + v_q i_q \\
Q = v_d i_q - v_q i_d
\end{cases}
\tag{10}
$$

On the other hand, Joule and iron losses are given by the following formulas:

$$
\begin{cases}
P_{cu} = R_s i_d{}^2 + R_s i_q{}^2 \\
P_{ir} = \dfrac{3}{2} N_r \Omega \left[ M_{dq} \left( i_d^2 - i_q^2 \right) - (L_d - L_q) i_d i_q \right] + \dfrac{1}{2} \left( L_d \dfrac{i_d^2}{dt} + L_q \dfrac{i_q^2}{dt} \right) + M_{dq} \dfrac{d(i_d i_q)}{dt}
\end{cases}
\tag{11}
$$

with $P_{em}$ represents DSPMG electromagnetic power, given by:

$$
P_{em} = \Omega\, T_{em}
\tag{12}
$$

The power factor can be evaluated with the help of the mean values of active and reactive powers as:

$$
\cos \psi = \left| \dfrac{P_{a-mean}}{\sqrt{P_{a-mean}^2 + Q_{mean}{}^2}} \right|
\tag{13}
$$

Parameters and variables that appear in Equation (1) to Equation (13) are defined in Table A1 (Appendix A).

In order to implement the OR−DSPMG model, state equation must be established. The basic state formula is defined as:

$$
\dot{x} = A\, x + B\, y
\tag{14}
$$

where:

$$
x = \left[ i_d, i_q, \Omega \right]^t, \quad y = \left[ (v_d + e_{md}), (v_q + e_{mq}), (T_{em} - T_m) \right]^t
\tag{15}
$$

Direct and quadrature magneto-mortice forces are expressed by:

$$
\begin{bmatrix} e_{md} \\ e_{mq} \end{bmatrix} = \sqrt{\dfrac{3}{2}}\, w_e\, \varphi_1 \begin{bmatrix} 0 \\ 1 \end{bmatrix}
\tag{16}
$$

Based on Equations (5)–(7) and (9), matrix A and B elements are defined as follows:

$$
A = \begin{bmatrix} a_{11} & a_{12} & 0 \\ a_{21} & a_{22} & 0 \\ 0 & 0 & -\frac{f_v}{J_t} \end{bmatrix}, \quad B = \begin{bmatrix} b_{11} & b_{12} & 0 \\ b_{21} & b_{22} & 0 \\ 0 & 0 & \frac{1}{J_t} \end{bmatrix}
\tag{17}
$$

with

$$\left\{ \begin{array}{l} a_{11} = -\dfrac{[R_s - w_e(L_1+2M_1)\sin 3\theta_e][2(L_0-M_0)-(L_1+2M_1)\cos 3\theta_e]+[w_e(L_1+2M_1)\sin 3\theta_e][L_0-M_0-(L_1+2M_1)\cos 3\theta_e]}{2(L_0-M_0)^2 - \frac{1}{2}(L_1+2M_1)^2} \\[3mm] a_{12} = -\dfrac{[R_s+w_e(L_1+2M_1)\sin 3\theta_e][(L_1+2M_1)\sin 3\theta_e]-[w_e(L_0-M_0)+(L_1+2M_1)\cos 3\theta_e][2(L_0-M_0)-(L_1+2M_1)\cos 3\theta_e]}{2(L_0-M_0)^2 - \frac{1}{2}(L_1+2M_1)^2} \\[3mm] a_{21} = -\dfrac{[R_s-w_e(L_1+2M_1)\sin 3\theta_e][(L_1+2M_1)\sin 3\theta_e]+[w_e(L_0-M_0)-(L_1+2M_1)\cos 3\theta_e][2(L_0-M_0)+(L_1+2M_1)\cos 3\theta_e]}{2(L_0-M_0)^2 - \frac{1}{2}(L_1+2M_1)^2} \\[3mm] a_{22} = -\dfrac{[R_s+w_e(L_1+2M_1)\sin 3\theta_e][2(L_0-M_0)+(L_1+2M_1)\cos 3\theta_e]-[w_e(L_1+2M_1)\sin 3\theta_e][L_0-M_0+(L_1+2M_1)\cos 3\theta_e]}{2(L_0-M_0)^2 - \frac{1}{2}(L_1+2M_1)^2} \end{array} \right. \tag{18}$$

and

$$\left\{ \begin{array}{l} b_{11} = -\dfrac{2(L_0-M_0)-(L_1+2M_1)\cos 3\theta_e}{2(L_0-M_0)^2 - \frac{1}{2}(L_1+2M_1)^2} \\[3mm] b_{12} = -\dfrac{(L_1+2M_1)\sin 3\theta_e}{2(L_0-M_0)^2 - \frac{1}{2}(L_1+2M_1)^2} \\[3mm] b_{21} = -\dfrac{(L_1+2M_1)\sin 3\theta_e}{2(L_0-M_0)^2 - \frac{1}{2}(L_1+2M_1)^2} \\[3mm] b_{22} = -\dfrac{2(L_0-M_0)+(L_1+2M_1)\cos 3\theta_e}{2(L_0-M_0)^2 - \frac{1}{2}(L_1+2M_1)^2} \end{array} \right. \tag{19}$$

## 3. WECS Control

### 3.1. OR–DSPMSG–Side Converter Control

The basic control scheme of the generator side converter is illustrated in Figure 3. The limit between the MTPA control strategy region and flux weakening region is constrained by current and voltage limits.

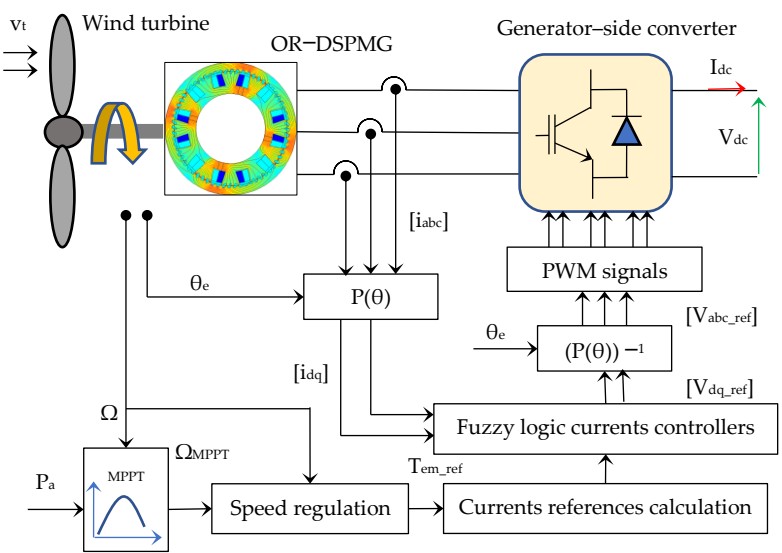

**Figure 3.** Generator–side converter control structure.

3.1.1. Maximum Torque Per Ampere Strategy

The current amplitude is given by:

$$I_a = \sqrt{\frac{2}{3}\left(i_d^2+i_q^2\right)} \tag{20}$$

Considering quasi sinusoidal currents wave forms, defined as follows:

$$\left\{ \begin{array}{l} i_a = -I_a(\theta_e)\sin(\theta_e+\theta_0) \\ i_b = -I_a(\theta_e)\sin(\theta_e-\frac{2\pi}{3}+\theta_0) \\ i_c = -I_a(\theta_e)\sin(\theta_e+\frac{2\pi}{3}+\theta_0) \end{array} \right. \tag{21}$$

electromagnetic torque can be expressed by:

$$T_{em} = -\left[\frac{3}{2}N_r\varphi_1\cos(\theta_0)\,I_a(\theta_e) + \frac{3}{8}N_r(L_1+2M_1)\sin(3\,\theta_e+2\theta_0)I_a^2(\theta_e)\right] \qquad (22)$$

then, current amplitude, depending on $\theta_e$ and load angle $\theta_0$, is given by:

$$I_a(\theta_e) = \frac{-\frac{3}{2}\,N_r\varphi_1\cos(\theta_0) + \sqrt{\left(\frac{3}{2}\,N_r\varphi_1\cos(\theta_0)\right)^2 - \frac{3}{2}\,N_r(L_1+2M_1)\sin(3\,\theta_e+2\theta_0)T_{em}}}{\frac{3}{4}\,N_r(L_1+2M_1)\sin(3\,\theta_e+2\theta_0)} \qquad (23)$$

However, in accordance with quadratic equation principle, Equation (23) is constrained by the following inequality:

$$(\cos(\theta_0))^2 \geq \frac{2(L_1+2M_1)T_{em}}{3N_r\varphi_1^2} \qquad (24)$$

Based on Equation (20), d–q currents references can be expressed as:

$$\begin{cases} i_d = -\sqrt{\frac{3}{2}}I_a\sin(\delta) \\ i_q = \sqrt{\frac{3}{2}}I_a\cos(\delta) \end{cases} \qquad (25)$$

By referring to Equation (8), electromagnetic torque can be expressed as:

$$T_{em} = -\sqrt{\frac{3}{2}}N_r\varphi_1 i_q + \frac{(L_1+2M_1)}{4}N_r\left[2\,i_d i_q\cos(3\,\theta_e) + \left(i_d^2 - i_q^2\right)\sin(3\,\theta_e)\right] \qquad (26)$$

Based on Equations (25) and (26), electromagnetic torque becomes:

$$T_{em} = -\frac{3}{2}N_r\varphi_1 I_a\cos(\delta) - \frac{3}{8}N_r(L_1+2M_1)\,I_a^2\sin(2\delta + 3\,\theta_e) \qquad (27)$$

Torque maximization must agree with the following condition:

$$\frac{\partial T_{em}}{\partial \delta} = 0 \qquad (28)$$

Consequently, the following equation is obtained:

$$(L_1+2M_1)I_a\left(\sin\left(\delta - \frac{3}{2}\theta_e\right)\right)^2 + \varphi_1\sin(\delta) - \frac{1}{2}(L_1+2M_1)I_a = 0 \qquad (29)$$

Solving Equation (29) allows to obtain a real solution given by the following formula:

$$\delta = \sin^{-1}\left(\frac{-\varphi_1 + \sqrt{\varphi_1^2 + 2(L_1+2\,M_1)^2 I_a^2}}{2\,I_a\,(L_1+2\,M_1)}\right) \qquad (30)$$

Generator MTPA control is applied between points o (0,0) and $A_1$($id_1$, $iq_1$) (Figure 4) and the corresponding currents references are given by Equation (25). From point $A_1$, the generator is controlled through flux weakening strategy; the corresponding speed limit will be determined in the next paragraph.

### 3.1.2. Flux Weakening Control

DC bus voltage, $V_{dc}$, is determined in the basis of maximal generator voltage, $V_{lim}$; consequently, the DC voltage must agree with the following inequality:

$$V_{dc} \geq 2\,V_{lim} \qquad (31)$$

On the other hand, voltage limit is defined by:

$$V_{lim} = \sqrt{\frac{2}{3}\left(v_d^2 + v_q^2\right)}\tag{32}$$

In order to simplify equation complexity, we consider only steady states and that stator resistances are neglected; thus, simplified voltage equations are expressed as follows:

$$\begin{cases} v_d' = -\left(2w_e M_{dq}\right)i_d + \frac{w_e}{2}(3L_d - L_q)i_q \\ v_q' = \left(2w_e M_{dq}\right)i_q - \frac{w_e}{2}(3L_q - L_d)i_d - \sqrt{\frac{3}{2}}\varphi_1 w_e \end{cases}\tag{33}$$

Based on Equations (32) and (33), the elliptic equation is obtained:

$$A\,i_d^2 + B\,i_d i_q + C\,i_q^2 + D\,i_d + E\,i_q + F = 0\tag{34}$$

where

$$\begin{cases} A = 4(L_0 - M_0)^2 + 4(L_1 + 2M_1)^2 - 8(L_0 - M_0)(L_1 + 2M_1)\cos(3\theta_e) \\ B = 16(L_0 - M_0)(L_1 + 2M_1)\sin(3\theta_e) \\ C = 4(L_0 - M_0)^2 + 4(L_1 + 2M_1)^2 + 8(L_0 - M_0)(L_1 + 2M_1)\cos(3\theta_e) \\ D = 4\sqrt{6}\varphi_1((L_0 - M_0) - (L_1 + 2M_1)\cos(3\theta_e)) \\ E = 4\sqrt{6}\varphi_1(L_1 + 2M_1)\sin(3\theta_e) \\ F = 6\left(\varphi_1^2 - \frac{v_{lim}^2}{we_e^2}\right) \end{cases}\tag{35}$$

with respect to the following condition:

$$-4\left(\left(3L_q - L_d\right)\left(3L_d - L_q\right) - 16\,M_{dq}^2\right)^2 < 0\tag{36}$$

Solving Equation (34) is a hard task because parameters A, B, C, D, and E are dependent on $\theta_e$. In order to obtain an acceptable simplification of this equation, mean values of parameters A, B, C, D, and E around periodic interval $[0,\ 3\theta_e]$ are taken into account. So, the expressions given by Equation (34) become:

$$\begin{cases} A = 4(L_0 - M_0)^2 + 4(L_1 + 2M_1)^2 \\ B = 0 \\ C = 4(L_0 - M_0)^2 + 4(L_1 + 2M_1)^2 \\ D = 4\sqrt{6}\varphi_1(L_0 - M_0) \\ E = 0 \\ F = 6\left(\varphi_1^2 - \frac{v_{lim}^2}{w_e^2}\right) \end{cases}\tag{37}$$

Consequently, Equation (34) becomes:

$$\left(i_d + \frac{\overline{D}}{2\,\overline{A}}\right)^2 + i_q^2 - \frac{\overline{D}^2}{4\,\overline{A}^2} + \frac{F}{\overline{A}} = 0\tag{38}$$

Equation (38) represents a circle of voltage limit; therefore, generator operation point shifting in d–q plan is shown in Figure 4.

Based on Equation (38), electrical speed, $w_e$, can be written as:

$$w_e = \frac{\sqrt{6}\,V_{lim}}{\sqrt{\overline{A}\left[\left(i_d + \frac{\overline{D}}{2\,\overline{A}}\right)^2 + i_q^2 - \frac{\overline{D}^2}{4\,\overline{A}^2} + \frac{6\varphi_1^2}{\overline{A}}\right]}}\tag{39}$$

When operation point $A_1$ is reached, currents can be expressed as:

$$\begin{cases} i_{d1} = -\sqrt{\frac{3}{2}}\, I_{max}\sin(\delta_1) \\ i_{q1} = \sqrt{\frac{3}{2}}\, I_{max}\,\cos(\delta_1) \end{cases} \tag{40}$$

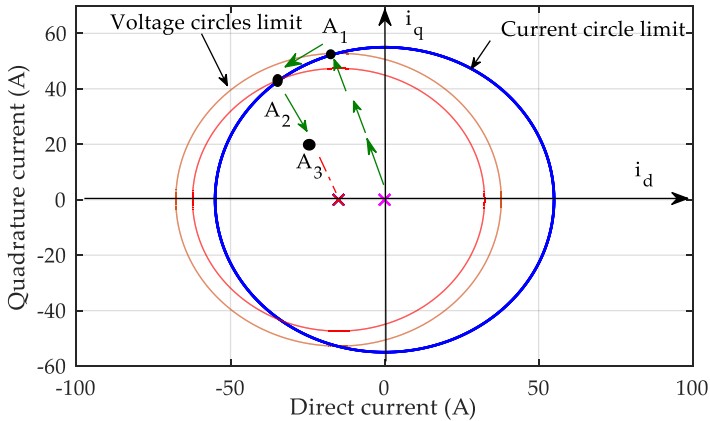

**Figure 4.** Generator operation trajectory in d–q current plan.

with

$$\delta_1 = \sin^{-1}\left(\frac{-\varphi_1 + \sqrt{\varphi_1^2 + 2(L_1 + 2\,M_1)^2 I_{max}^2}}{2\,I_{max}\,(L_1 + 2\,M_1)}\right) \tag{41}$$

Therefore, speed limit at point A1 is given by:

$$w_{e1} = \frac{\sqrt{6}\,V_{lim}}{\sqrt{\overline{A}\left[\left(i_{d1} + \frac{\overline{D}}{2\,\overline{A}}\right)^2 + i_{q1}^2 - \frac{\overline{D}^2}{4\,\overline{A}^2} + \frac{6\varphi_1^2}{\overline{A}}\right]}} \tag{42}$$

Between points $A_1$ and $A_2$, operation point, corresponding to first flux wakening area, is given by current and voltage circle limits intersection; relative equations to this zone are given by:

$$\begin{cases} i_d = -\sqrt{\frac{3}{2}}I_{max}\,\sin(\alpha) \\ i_q = \sqrt{\frac{3}{2}}I_{max}\cos(\alpha) \end{cases} \tag{43}$$

with

$$I_{max} = \sqrt{\frac{2}{3}\left(i_d^2 + i_q^2\right)} \tag{44}$$

Thus, according to Equations (38), (43), and (44), the following formula is obtained:

$$\alpha = \sin^{-1}\left(\frac{\frac{3}{2}\,\overline{A}I_{max}^2 + 6\left(\varphi_1^2 - \frac{V_{lim}^2}{w_e^2}\right)}{\sqrt{\frac{3}{2}}\,\overline{D}\,I_{max}}\right) \tag{45}$$

In order to agree with current and voltage limitation constraints, currents references must be recalculated from operation point $A_2$. Thus, according to Equation (38), direct current can be expressed as:

$$i_d = -\frac{\overline{D}}{2\,\overline{A}} - \Delta i_d \tag{46}$$

So, q–axis current can be written as:

$$i_q = \sqrt{\left| \frac{\overline{D}^2}{4\,\overline{A}^2} - \frac{6}{\overline{A}} \left( \varphi_1^2 - \frac{V_{lim}^2}{w_e^2} \right) - \Delta i_d^2 \right|} \tag{47}$$

Based on Equation (12) and according to Equations (26), (46), and (47), electromagnetic power formula is given by:

$$P_{em} = \mu_1 \sqrt{\frac{\overline{D}^2}{4\,\overline{A}^2} - \frac{F}{A} - \Delta i_d^2} + \mu_2 \left[ 2\left( \frac{\overline{D}}{2\,\overline{A}} + \Delta i_d \right) \sqrt{\frac{\overline{D}^2}{4\,\overline{A}^2} - \frac{F}{A} - \Delta i_d^2} \cos(3\theta_e) + \left( 2\,\Delta i_d^2 + \frac{\overline{D}}{\overline{A}}\Delta i_d + \frac{F}{A} \right) \sin(3\theta_e) \right] \tag{48}$$

with

$$\begin{cases} \mu_1 = \sqrt{\frac{3}{2}}\,\varphi_1 w_e \\ \mu_2 = \frac{(L_1 + 2M_1)}{4} w_e \end{cases} \tag{49}$$

In order to keep constant active power, the following condition must be verified:

$$\frac{\partial P_{em}}{\partial \Delta i_d} = 0 \tag{50}$$

Thus, the next equation is obtained:

$$\frac{(L_1 + 2M_1)}{4} \left[ \left( -4\,\Delta i_d^2 - \frac{\overline{D}}{\overline{A}}\Delta i_d - 2\frac{F}{A} + \frac{\overline{D}^2}{2\,\overline{A}^2} \right) \cos(3\theta_e) + \sqrt{\frac{\overline{D}^2}{4\,\overline{A}^2} - \frac{F}{A} - \Delta i_d^2} \left( 4\Delta i_d + \frac{\overline{D}}{\overline{A}} \right) \sin(3\theta_e) \right] - \sqrt{\frac{3}{2}}\,\varphi_1 \Delta i_d = 0 \tag{51}$$

Therefore, according to Equation (51), the accepted real solution is given by:

$$\Delta i_d = \frac{-\left( \sqrt{\frac{3}{2}}\,\varphi_1 + \frac{(L_1+2M_1)\,\overline{D}}{4\,\overline{A}} \right) + \sqrt{\left( \sqrt{\frac{3}{2}}\,\varphi_1 + \frac{(L_1+2M_1)\,\overline{D}}{4\,\overline{A}} \right)^2 + 4\,\frac{(L_1+2M_1)^2}{\overline{A}} \left( \frac{\overline{D}^2}{8\,\overline{A}} - 3\left( \varphi_1^2 - \frac{V_{lim}^2}{w_e^2} \right) \right)}}{2(L_1+2M_1)} \tag{52}$$

According to Equations (38) and (46) applied in the point $A_2$, the following expression is obtained:

$$w_{e2} = \frac{\sqrt{6}\,V_{lim}}{\sqrt{\frac{3}{2}\,\overline{A}\,I_{max}^2 + 6\,\varphi_1^2 - \frac{\overline{D}^2}{2\,\overline{A}} - \overline{D}\,\Delta i_{d2}}} \tag{53}$$

with

$$\Delta i_{d2} = \frac{-\left( \sqrt{\frac{3}{2}}\,\varphi_1 + \frac{3(L_1+2M_1)\,\overline{D}}{4\,\overline{A}} \right) + \sqrt{\left( \sqrt{\frac{3}{2}}\,\varphi_1 + \frac{3(L_1+2M_1)\,\overline{D}}{4\,\overline{A}} \right)^2 + (L_1+2M_1)^2 \left( 3I_{max}^2 - \frac{\overline{D}^2}{2\,\overline{A}^2} \right)}}{2(L_1+2M_1)} \tag{54}$$

From point $A_2$ ($id_2$,$iq_2$), the operation trajectory converge to point $A_3$ $(-\frac{\overline{D}}{2\,\overline{A}}, 0)$. Main parameters employed in this section are defined in Table A2 (Appendix A).

### 3.1.3. Fuzzy Logic Controller (FLC)

In order to obtain an accurate and robust control of generator d–q currents and DC voltage in both normal and flux weakening operating regions, fuzzy logic controllers are developed and employed instead of PI regulators.

The error between the set point value and the measured one of generator d–q currents ($i_d$, $i_q$) and DC voltage ($V_{dc}$) is processed through FLC for each as shown in Figure 5.

For the fuzzification step, seven fuzzy sets (NB, NM, NS, Z, PS, PM, PB) are used for the two input variables (E and dE). Output variable (dU) is defined with nine fuzzy sets (NVB, NB, NM, NS, Z, PS, PM, PB, PVB). Furthermore, tuning coefficients ($k_e$, $k_{de}$ and $k_{du}$)

are associated for inputs and output signals. Letters V, N, P, B, M, S, Z, and mean very, negative, positive, big, medium, small, and zero, respectively.

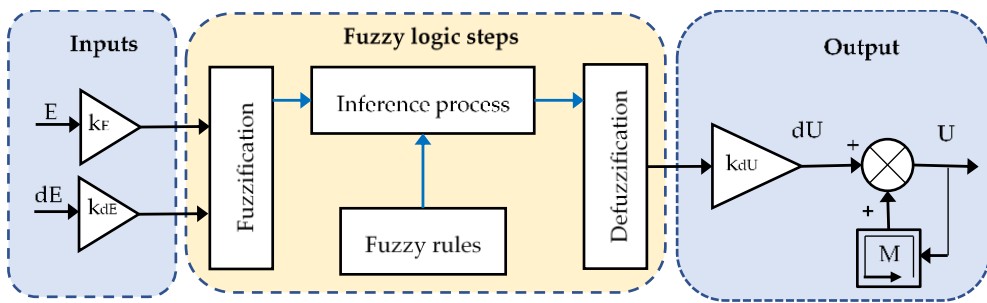

**Figure 5.** Basic scheme of fuzzy logic controller.

Triangular membership functions are employed for the fuzzification of error and error variation; however, defuzzification is addressed with the center of gravity method (Figure 6). After the defuzzification step, control signal is defined as follows:

$$U_k = U_{k-1} + dU_k \qquad (55)$$

Control rules table is built based on the characteristic of the step response: a large control signal is needed when the output is falling forward from the reference; however, while the output is near the reference, a small variation increase is required. Based on the previous reasoning, fuzzy rules table is obtained as given by Table 1.

**Table 1.** Fuzzy rules table.

| Output Signal (dU) | | Error (E) | | | | | | |
|---|---|---|---|---|---|---|---|---|
| | | **NB** | **NM** | **NS** | **Z** | **PS** | **PM** | **PB** |
| **Change of error (dE)** | NB | NVB | NVB | NVB | NB | NM | NS | Z |
| | NM | NVB | NVB | NB | NM | NS | Z | PS |
| | NS | NVB | NB | NM | NS | Z | PS | PM |
| | Z | NB | NM | NS | Z | PS | PM | PB |
| | PS | NM | NS | Z | PS | PM | PB | PVB |
| | PM | NS | Z | PS | PM | PB | PVB | PVB |
| | PB | Z | PS | PM | PB | PVB | PVB | PVB |

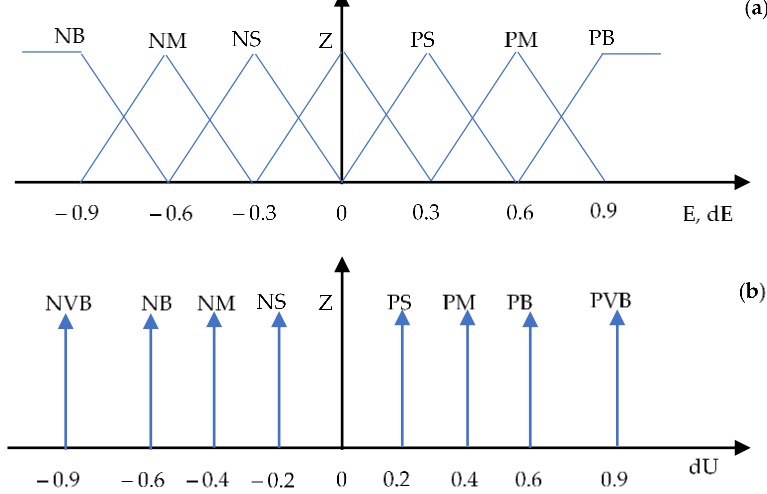

**Figure 6.** Fuzzy sets, (**a**): Input variables; (**b**): Output variable.

*3.2. Grid–Side Inverter Control*

The basic representation of the grid-side inverter control scheme is shown in Figure 7.

Electrical equation relative to DC bus voltage is given by:

$$\frac{dV_{dc}}{dt} = \frac{1}{C}(I_{dc} - I_{inv})$$ (56)

So, DC bus voltage regulation can be achieved using the following formula [31]:

$$I_{inv\_ref} = I_{dc} - FLC(V_{dc\_ref} - V_{dc})$$ (57)

where $I_{inv-ref}$ and $V_{dc-ref}$ are inverter reference current and DC bus voltage reference, respectively.

Considering a three-phase balanced system, grid active and reactive powers in d–q reference frame are expressed as follows:

$$\begin{cases} P_g = v_{gd}i_{gd} + v_{gq}i_{gq} \\ Q_g = v_{gq}i_{gd} - v_{gd}i_{gq} \end{cases}$$ (58)

Thus, grid currents references are expressed by:

$$\begin{bmatrix} i_{gd\_ref} \\ i_{gq\_ref} \end{bmatrix} = \frac{1}{v_{gd}^2 + v_{gq}^2} \begin{bmatrix} P_{g\_ref} & -Q_{g\_ref} \\ Q_{g\_ref} & P_{g\_ref} \end{bmatrix} \begin{bmatrix} v_{gd} \\ v_{gq} \end{bmatrix}$$ (59)

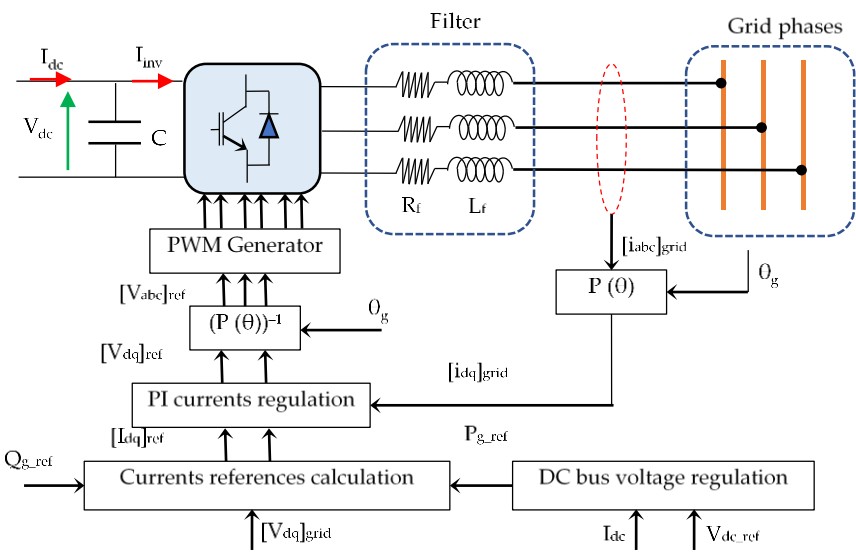

**Figure 7.** Grid–side inverter control strategy representation.

In order to operate with unitary power factor, reactive power reference is set to zero; therefore, active and reactive powers references are given by:

$$\begin{cases} P_{g\_ref} = P_{dc} = V_{dc}I_{dc} \\ Q_{g\_ref} = 0 \end{cases}$$ (60)

Currents regulations are insured by proportional-integral (PI) controllers [31]; controllers parameters (proportional gain $k_p$ and time constant $\tau_d$) are given in Table A2 (Appendix A).

## 4. Simulation Results and Discussion

A fixed pitch 10 kW WECS equipped with OR−DSPMG and grid−connected is modeled and implemented in Matlab/Simulink software. The system operates under MPPT control associated to MTPA strategy in partial load region. When the rated speed is reached ($\Omega_1$ = 4.92 rad/s), extracted power is limited by operating DSMPG under flux weakening

and torque control. Generator d–q currents in all operation regions are regulated by means of FLC, which is the same for DC voltage. However, grid-side d–q currents are regulated by means of PI correctors.

Full power back to back inverter and converter are controlled through pulse width modulation signals generated classically by intersection of a high frequency triangular carrier with voltages references. This modulation is characterized with frequency and maximal amplitude ratio of 10 kHz and 0.8, respectively. Inverter and converter losses are not taken into account in this study.

Obtained simulation results are presented in Figures 8–17.

### 4.1. Wind Turbine

Wind speed profile covers all operating regions (partial load and full load system operation) as shown in Figure 8. In the partial load region, MPPT control allows keeping the power coefficient and tip speed ratio at their maximum and optimal values ($C_{pmax} = 0.437$, $\lambda_{opt} = 2.41$), respectively. However, when the nominal regime is reached ($v_n = 8.70$ m/s, $P_n = 10$ kW), the power coefficient decreases while the tip speed ratio increases (Figure 9); this corresponds to wind turbine operation in the right side of the $C_p$ ($\lambda$) curve as previously mentioned in Figure 2a.

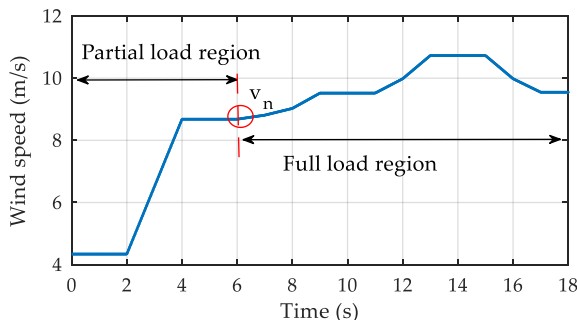

**Figure 8.** Wind speed.

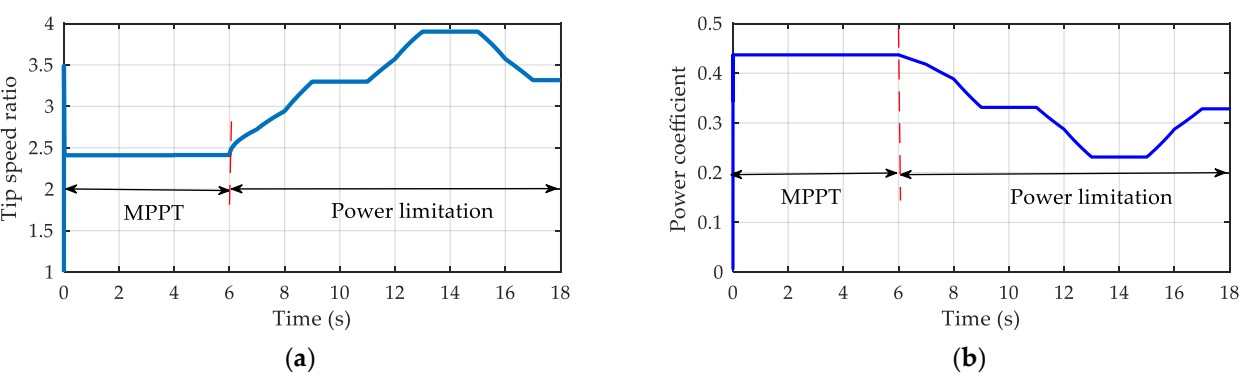

**(a)**                                                                 **(b)**

**Figure 9.** Wind turbine characteristics: (**a**) Tip speed ratio; (**b**) Power coefficient.

### 4.2. OR−DSPMG

Generator mechanical speed, d–q currents, and magnetic flux are shown in Figure 10. OR−DSPMG operation is divided into three regions: below speed $\Omega_1$, between speeds $\Omega_1$ and $\Omega_2$, and above speed $\Omega_2$ ($\Omega_1 = 4.92$ rad/s and $\Omega_2 = 5.43$ rad/s). In the first region, the extracted power is maximized by MPPT strategy associated to MTPA for torque control. Maximum torque trajectory in the d–q plane is described from point (0,0) to point $A_1$ (Figure 10c). Currents and flux values corresponding to 0.5 $\Omega_1$ and rated speed $\Omega_1$ are given in Table 2. A maximum quadrature current and flux of 54.5 A and 2.06 Wb are obtained at point $A_1$. With negative direct current, direct flux decreases until approaching 0 at the point $A_1$.

**Table 2.** Numerical values of d–q currents and flux for 50% $\Omega_1$ and $\Omega_1$.

| Time (s) | $\Omega$ (rad/s) | $i_d$ (A) | $i_q$ (A) | $\Phi_d$ (Wb) | $\Phi_q$ (Wb) |
|:---:|:---:|:---:|:---:|:---:|:---:|
| 1 | 2.453 | −0.95 | 13.52 | 0.512 | 0.553 |
| 5 | 4.92 | −14.3 | 54.5 | 0.05 | 2.06 |

Above speed limit $\Omega_1$, the system operates in the overspeed region; therefore, between speeds $\Omega_1$ and $\Omega_2$, corresponding to the first flux weakening region, sinusoidal currents waveform is imposed with respect to current and voltage limitations. In fact, the operating point moves on maximum current amplitude circle from point $A_1$ to point $A_2$ (Figure 10c). In order to agree with voltage and power limitations, above speed limit $\Omega_2$, OR−DSPMG operate under flux weakening control with new operating point trajectory, described by section $A_2$–$A_3$ in Figure 10c. Under flux weakening operation, quadrature current decreases while absolute d–axis current value increases, which is accompanied by q–axis flux decreasing and absolute d–axis flux value increasing. In fact, direct flux becomes a reluctant flux (demagnetizing) and the total machine flux is weakened, which allows overspeed operations.

Direct and quadrature currents presented in Figure 10b show a small ripple; in fact, maximum ripples are observed in the overspeed region with 0.26 A (2.78%) and 0.23 A (1.77%) for d–axis and q–axis currents, respectively. Furthermore, fuzzy logic controllers show satisfactory performances (precision, time response, and overshoot).

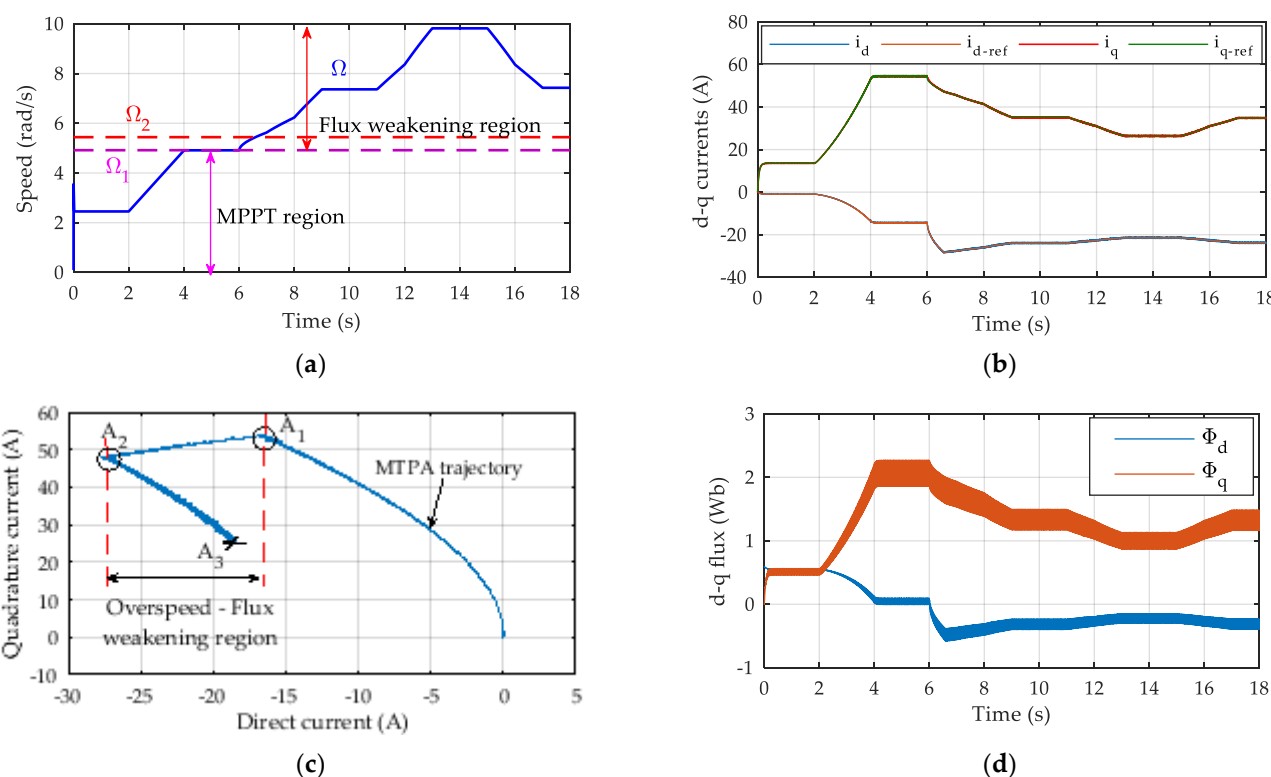

**Figure 10.** OR−DSPMG speed, currents, and flux: (**a**) Mechanical speed; (**b**) Direct and quadrature currents; (**c**) Current amplitude; (**d**) Direct and quadrature flux.

Active and reactive power and Joule loss of the OR–DSMPG are presented in Figure 11. From 0 to 6 s, corresponding to the power maximization region, a 10.23 kW mean active power is achieved when the rated speed $\Omega_1$ is reached. However, reactive power increases rapidly and reaches a mean value of 34.73 kVAR, which greatly affects the machine power factor (presented in Figure 12) which reaches to a value of 0.28 at the rated speed. Indeed, a great amount of reactive power absorbed by the OR−DSPMG is necessary for machine magnetization and torque production. In the overspeed region, active power is limited

to approximately 9.7 kW, while the reactive power and Joule decrease attended with improvement of power factor. For particular values of speed (50% $\Omega_1$, 100% $\Omega_1$, 150% $\Omega_1$, and 200% $\Omega_1$), average active and reactive powers, Joule loss, and power factor values are given in Table 3.

**Table 3.** Torque, powers, and power factor for particular speed values.

| Speed (rad/s) | $0.5 \times \Omega_1$ | $\Omega_1$ | $1.5 \times \Omega_1$ | $2.5 \times \Omega_1$ |
|---|---|---|---|---|
| Current amplitude (A) | 11.1 | 46 | 33.25 | 26.2 |
| Mean torque value (N.m) | −510 | −2025 | −1298 | −978 |
| Torque ripples (%) | 8.37% | 37.5% | 29.35% | 23.95% |
| Mean active power value (kW) | − 1.265 | −10.230 | −9.705 | −9.680 |
| Joule losses (kW) | 0.016 | 0.281 | 0.146 | 0.091 |
| Mean Reactive power (kVAR) | 0.989 | 34.73 | 23.61 | 17.55 |
| Power factor | 0.79 | 0.28 | 0.38 | 0.48 |

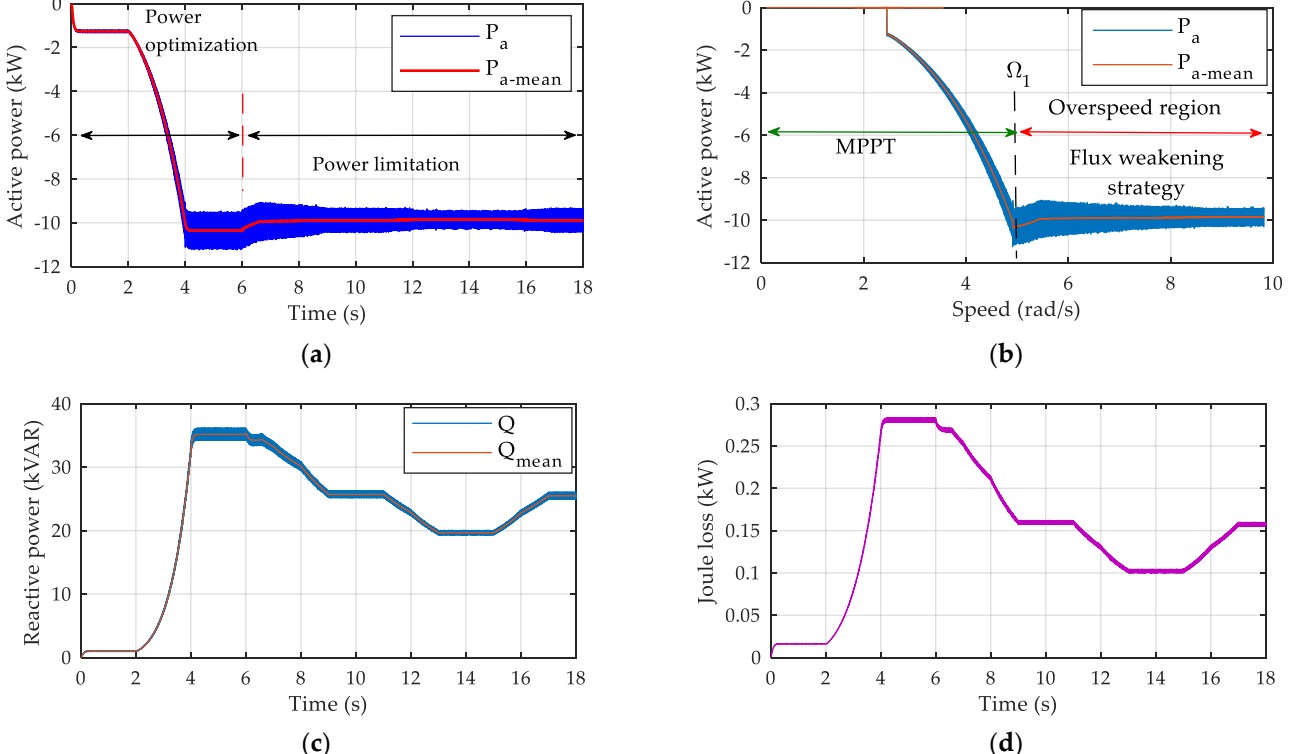

**Figure 11.** OR−DSPMG actives and reactive powers: (**a**) Active power; (**b**) Active power versus speed; (**c**) Reactive power; (**d**) Joule loss.

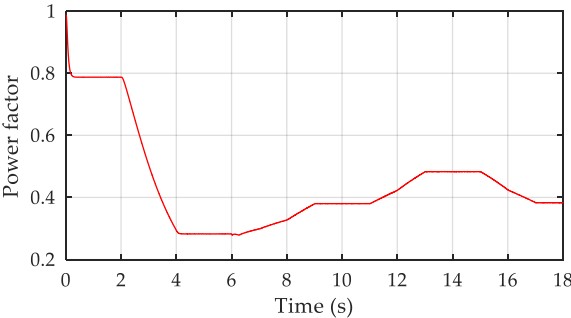

**Figure 12.** OR−DSPMG power factor.

Electromagnetic and mechanical torques and average electromagnetic torque vs. speed are presented in Figure 13. From 0 to 6 s (corresponding to the rated speed $\Omega_1$), electromagnetic torque increases along maximum torque trajectory; thus, a maximum average value of 2025 Nm is reached with maximum ripples of 37.5% (see the zoom in Figure 13b). Electromagnetic torque ripple coefficient $C_r$ is evaluated as follows:

$$C_r = \left| \frac{T_{em-max} - T_{em-min}}{T_{em-mean}} \right| \qquad (61)$$

where $T_{em\text{-}mean}$, $T_{em\text{-}max}$ and $T_{em\text{-}min}$ are the average electromagnetic torque value, the maximum torque, and the minimum torque, respectively.

Torque ripples are mainly caused by the OR−DSPMG mutual inductances $M_0$ and $M_1$. In the overrated speed region, the decrease of the current amplitude is attended with electromagnetic torque and this ripple coefficient decreases. Values of current amplitude, average electromagnetic torque, and torque ripple coefficient corresponding to particular speed values are also given in Table 3.

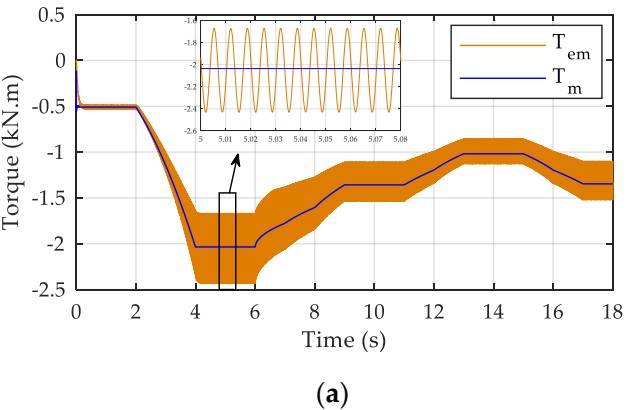
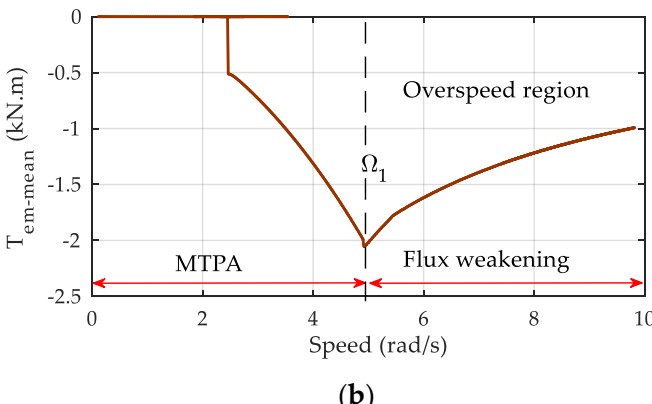

**(a)** **(b)**

**Figure 13.** OR−DSPMG torque: (**a**) Electromagnetic and mechanical torque; (**b**) Mean electromagnetic torque versus speed.

OR−DSPMG phase current and voltage wave forms and their FFT analysis for 50 Hz are presented in Figure 14. Since the DC-bus voltage is equal to 1200 V, the maximum voltage and current amplitudes are, respectively, 800 V and 46 A with 50 Hz frequency. However, voltage and current waveforms are affected by a THD of 21% and 0.55%, respectively; furthermore, the fundamental component for both voltage and current are 525.4 V and 45.98 A, respectively. As can be seen, the voltage waveforms are affected by the second harmonic, mainly caused by the machine mutual inductances $M_0$ and $M_1$.

Table 4 provides amplitude of current and voltage relative to particular fundamental frequencies: 25 Hz (time = 1.5 s), 50 Hz (time = 5 s), 75 Hz (time = 10 s), and 100 Hz (time = 14 s). Only pertinent harmonic orders (2nd and 4th) are taken into account. As can be noticed in Table 4, for all signal frequencies, the second harmonic affects the voltage wave form; thus, the maximum THD is observed for 100 Hz signal frequency (27.14%).

### 4.3. DC Bus and Grid

Grid side converter control aims to control and maintain the DC bus voltage constant and assure active power transmission to the grid side with unitary power factor. Figure 15 presents the DC bus voltage with a set point value of 1200 V. As can be observed, there is a small ripple in the voltage, with maximum ripple equal to 0.01 V; consequently, the fuzzy logic controller demonstrates an excellent precision and time response.

Inverter voltage and line current with their FFT analysis are shown in Figure 16. Since the DC bus voltage is equal to 1200 V and the grid voltage RMS value, Vg_rms, is equal to 690 V, the modulated inverter voltage and line current amplitude are, respectively, equal to

800 V and 7 A. Voltage and current are not greatly affected by THD; only 1.95% and 0.55% of THD are observed for voltage and current, respectively.

**Table 4.** OR–DSMPSG FFT analysis of phase voltage and current.

| Base Frequency | Harmonic Order | Voltage (V) | Current (A) |
|:---:|:---:|:---:|:---:|
| 25 Hz | Fundamental (25 Hz)<br>2nd harmonic (50 Hz)<br>4th harmonic (100 Hz) | 96.48 (100%)<br>13.17 (13.65%)<br>0.45 (0.47%) | 11.09 (100%)<br>0.024 (0.22%)<br>0.006 (0.06%) |
| 50 Hz | Fundamental (50 Hz)<br>2nd harmonic (100 Hz)<br>4th harmonic (200 Hz) | 525.4 (100%)<br>109.33 (20.81%)<br>6.83 (1.3%) | 45.98 (100%)<br>0.15 (0.33%)<br>0.06 (0.14%) |
| 75 Hz | Fundamental (75 Hz)<br>2nd harmonic (150 Hz)<br>4th harmonic (300 Hz) | 516.4 (100%)<br>124.92 (24.19%)<br>4.28 (0.83%) | 33.22 (100%)<br>0.14 (0.44%)<br>0.03 (0.10%) |
| 100 Hz | Fundamental (100 Hz)<br>2nd harmonic (200 Hz)<br>4th harmonic (400 Hz) | 506.9 (100%)<br>137.57 (27.14%)<br>12.92 (2.55%) | 26.16 (100%)<br>0.15 (0.60%)<br>0.04 (0.16%) |

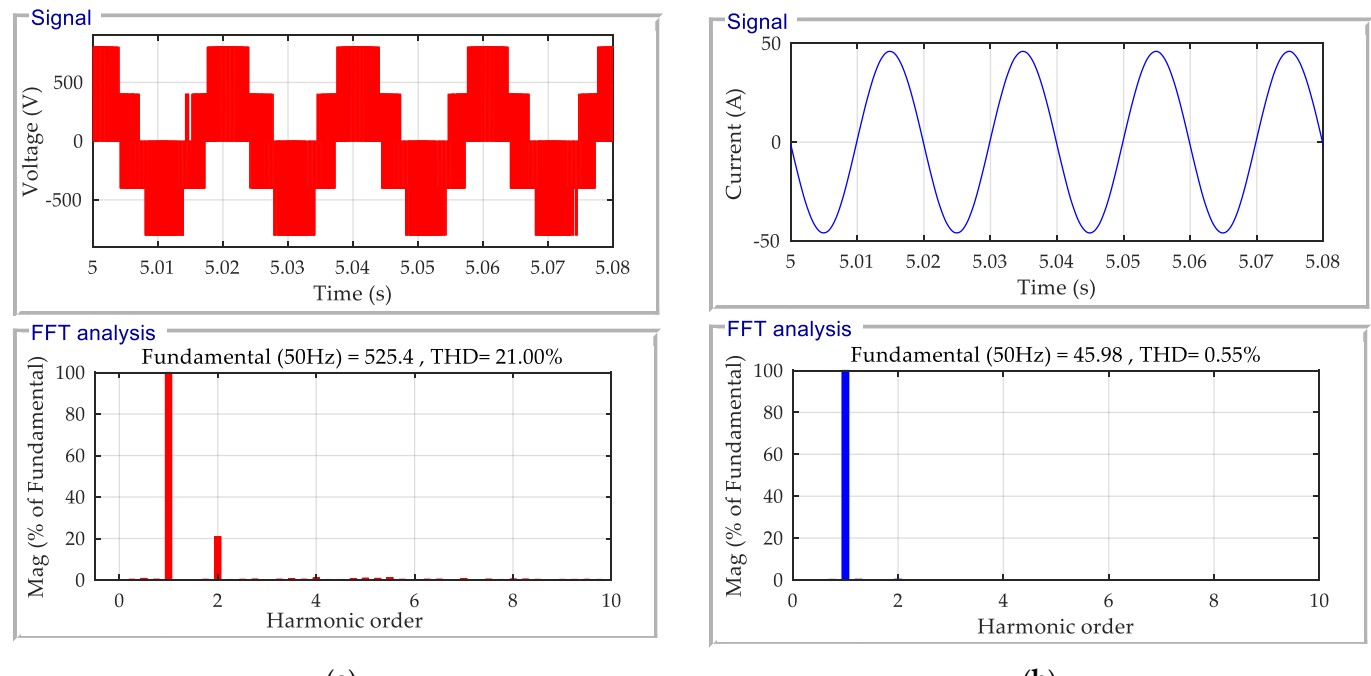

(**a**)             (**b**)

**Figure 14.** OR−DSPMG phase voltage and current for 50 Hz: (**a**) Voltage wave form and FFT analysis (**b**) Current wave form and FFT analysis.

Figure 17 presents active and reactive power exchanged with the electrical grid. Initially, between [0, 2] s, transmitted active power is equal to 2.5 kW; in the rated zone, [4, 6] s, a value of 10.2 kw is sent to grid. In the overspeed region [0, 18] s, active power is still approximately constant at 9.7 kW (inverter loss are not taken into account). As can be seen, the reactive power is equal to zero for all operation regions, which gives unitary power factor (ideal electrical grid is considered).

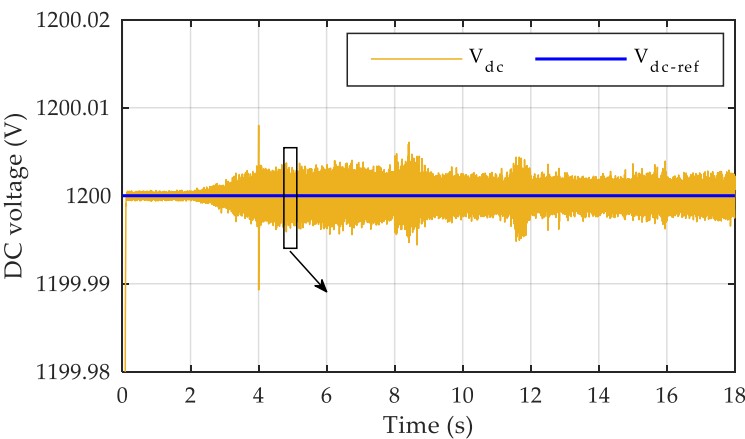

**Figure 15.** DC voltage.

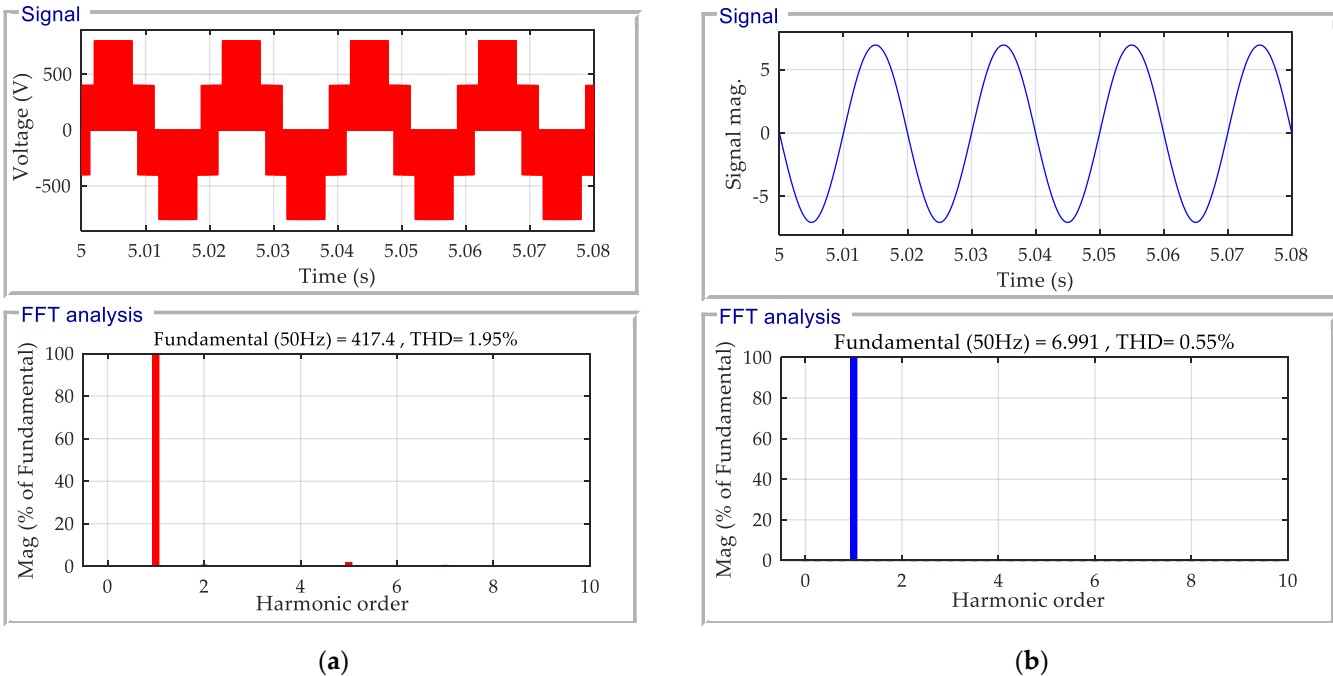

(**a**)                                                                                                         (**b**)

**Figure 16.** Grid phase voltage and current for 50 Hz: (**a**) Voltage wave form and FFT analysis (**b**) Current wave form and FFT analysis.

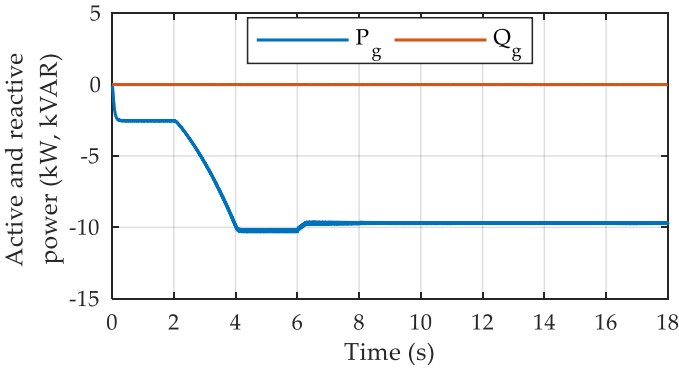

**Figure 17.** Grid active and reactive powers.

## 5. Conclusions

In this paper, the analysis of dynamic performance of a grid–connected WECS based direct drive OR−DSPMG is achieved. The main objective was to ensure system operation in the overspeed region with power limitation without pitch turbine blades control. Below the rated speed the MTPA strategy allows machine operation under maximum torque trajectory by mean of d–q currents regulations. As a result, a maximum average torque of 2025 Nm is obtained. Above the rated speed (4.92 rad/s), machine flux weakening control strategy was imposed in two steps: firstly, operation point was moved on current circle limit until the speed limit of this region is reached (5.44 rad/s). Secondly, flux weakening trajectory was modified in order to agree with power and voltage limitations; consequently, an overspeed of two times the rated speed is reached (9.84 rad/s). For all defined operating regions, corresponding d–q currents are calculated and regulated by means of fuzzy logic controllers, allowing to obtain satisfactory regulation performances in precision, time response, settling time, and overshoot, regardless of system settings. Similar regulation performances were obtained for DC voltage regulation, which contribute with a grid side converter control strategy to obtain an acceptable voltage and current wave form with very small THD. Furthermore, active power was transmitted to the grid with unitary power factor. However, system robustness against parameter variations is not studied in this work. An interesting future focus will be the development of an experimental platform in order to confirm the simulation results.

**Author Contributions:** Conceptualization, A.R., C.G. and J.-F.C.; methodology, A.R. and C.G. software, validation, A.R., C.G. and J.-F.C.; formal analysis, investigation, resources, data curation, A.R., C.G. and J.-F.C.; writing—original draft preparation, A.R., C.G. and J.-F.C.; writing—review and editing, A.R., C.G., J.-F.C., Y.L.K., T.K. and N.B.; visualization, A.R., C.G. and J.-F.C.; supervision, A.R., C.G. and J.-F.C. All authors have read and agreed to the published version of the manuscript.

**Funding:** This research received no external funding.

**Data Availability Statement:** Not applicable.

**Conflicts of Interest:** The authors declare no conflict of interest.

## Appendix A

Variables and parameters associated to wind turbine, OR−DSPMG, DC bus and grid are presented in Table A1

**Table A1.** Variables and parameters of wind turbine, OR−DSPMG, DC bus and grid.

| | Variables | Parameters |
|---|---|---|
| Wind turbine | $v_t$: wind speed (m/s) <br> $C_p$: power coefficient <br> $\lambda$: tip speed ratio <br> $T_m$: mechanical torque (Nm) <br> $P_t$: aerodynamic power (kW) <br> $P_{t\text{-}max}$: maximum extracted power (kW) <br> $\Omega$: mechanical speed (rpm) <br> $\Omega_{MPPT}$: optimal mechanical speed (rad/s) | $\rho$: air density (1.225 kg/m$^3$) <br> R: wind turbine radius (4.2633 m) <br> $C_{pmax}$: maximum power coefficient (0.4369) <br> $\lambda_{opt}$: optimal tip speed ratio (2.41) <br> $v_n$: rated wind speed (8.70 m/s) <br> $P_n$: rated wind turbine power (10 kW) <br> $v_{cut\text{-}in}$: cut-in speed (2.5 m/s) <br> Number of blade: 3 |

**Table A1.** *Cont.*

| | Variables | Parameters |
|---|---|---|
| OR–DSPMSG | $P(\theta_e)$: Park transformation matrix<br>$v_d$, $v_q$: direct and quadrature voltages (V)<br>$i_d$, $i_q$: direct and quadrature currents (A)<br>$\varphi_d$, $\varphi_q$: direct and quadrature flux (A)<br>$e_{md}$: direct magneto-mortice forces (V)<br>$e_{md}$: quadrature magneto–mortice forces (V)<br>$w_e$: electrical velocity (rad/s)<br>$\theta_e$: electrical position (rad)<br>$L_d$, $L_q$: direct and quadrature inductance (H)<br>$M_{dq}$: mutual inductance (H)<br>$T_{em}$: electromagnetic torque (Nm)<br>$T_{em\text{-}mean}$: average torque value<br>$T_{em\text{-}max}$: maximum torque<br>$T_{em\text{-}min}$: minimum torque<br>$C_r$: torque ripples<br>$P_{em}$: electromagnetic power (kW)<br>$P_a$: active power (kW)<br>$Q$: reactive power<br>$P_{cu}$, Joule loss (kW)<br>$P_{ir}$: iron losses (kW)<br>$P_{a\text{-}mean}$: mean active power (kW)<br>$Q_{mean}$: mean reactive power (kVAR)<br>$\cos\psi$: power factor<br>$I_a(\theta_e)$: current amplitude (A) | $R_s$: stator resistance (88.37 mΩ)<br>$L_0$: continuous self-inductance (25.5 mH)<br>$L_1$: first harmonic self-inductance (2.5 mH)<br>$M_0$: continuous mutual inductance ($-12.4$ mH)<br>$M_1$: first harmonic mutual inductance (2.5 mH)<br>$\varphi_1$: first harmonic PM flux (0.4805 Wb)<br>$N_r$: number of teeth in the rotor (64)<br>$J_t$: rotor inertia (Nm s$^{-1}$)<br>$f_v$: viscous friction coefficient<br>$B_r$: PMs magnetization (1.29 T)<br>$\mu_r$: relative permeability (1.049)<br>$E_m$: magnet thickness (27.05 mm)<br>$N_s$: stator teeth (48)<br>$N_r$: rotor teeth (64)<br>$N_{ps}$: stator pole (12)<br>hs: stator teeth depth (7.70 mm)<br>hr: rotor teeth depth (7.70 mm)<br>$E_s$: stator yoke thickness (40.90 mm)<br>$E_r$: rotor yoke thickness (29.65 mm)<br>$R_{r\text{-}out}$: outer rotor radius (300 mm)<br>$R_a$: slot radius (218.5 mm)<br>L: axial length (200 mm)<br>g: air gap thickness (0.5 mm)<br>M: active masse (184.4 kg)<br>$T_{out}$: average output torque (3504 Nm)<br>n: rated speed (50 rpm) |
| DC bus and electrical grid | $V_{dc}$: DC bus voltage (V)<br>$I_{dc}$: converter–side current (A)<br>$I_{inv}$: inverter side current (A)<br>$P_{dc}$: DC bus active power (kW)<br>$i_{gd}$, $v_{gd}$: grid direct current and voltage (A, V)<br>$i_{gq}$, $v_{gq}$: grid quadrature current and voltage (A, V)<br>$\theta_g$: electrical grid angle (rad)<br>$P_{g}$: grid active power (kW)<br>$Q_g$: grid reactive power (kVAR) | $V_{dc\text{-}ref}$: DC bus voltage reference (1200 V)<br>C: DC capacitance ($8 \times 10^{-4}$ F)<br>$R_f$: filter resistance (0.001 Ω)<br>$L_f$: filter inductance ($L_f$)<br>$V_{g\text{-}rms}$: voltage RMS value (690 V)<br>$k_p$: proportional regulator gain (10)<br>$\tau_d$: time constant ($1 \times 10^{-3}$ s) |

Parameters used in MTPA, flux weakening and fuzzy logic control strategies are summarized in Table A2.

**Table A2.** Main parameters of MTPA, flux weakening and fuzzy logic controllers.

| | Definition | Numerical Value |
|---|---|---|
| MTPA and flux weakening | Load angle ($\theta_0$)<br>Voltage limit ($V_{lim}$)<br>Maximum current ($I_{max}$)<br>Speed limit $\Omega_1$<br>Speed limit $\Omega_2$<br>Constant: $\overline{A}$<br>Constant: $\overline{D}$ | $\pi/72$<br>526 V<br>45 A<br>4.9218 rad/s<br>5.4375 rad/s<br>0.006<br>0.1784 |
| Fuzzy logic controller | Coefficient: $k_e$<br>Coefficient: $k_{de}$<br>Coefficient: $k_{du}$ | 0.1<br>10<br>100 |

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
