# Peer review of "Control of an Outer Rotor Doubly Salient Permanent Magnet Generator for Fixed Pitch kW Range Wind Turbine Using Overspeed Flux Weakening Operations"

_actuators, doi:10.3390/act12040168_

Round 1

Reviewer 1 Report

This paper describes with the analysis of the dynamic performance of a generator with a doubly salient exterior rotor excited by permanent magnets. It is well written and informative for the reader.

Some improvements are recommended:

1) Title kilo-Watt must be with small “k”. “flux weakening” don’t sounds relevant. “flux reluctance” is an alternative.

2) Define symbols for all equations. Appendix A does not cover all.

3) Sized drawing or size values of the generator is needed. Provide properties for permanent magnets and steels. Same for the wind turbine, I don’t find these values in “Table A2. Main parameters of WECS”.

4) Figure 16 (a)-(b) don’t seems to be related. It is completely similar with Figure 14.

5) Experimental verification is not included?

6) Deeper Discussion on the results is expected.

7) Enlarge Conclusion please.

Author Response

The authors thank the reviewer for the valuable comments. We have considered all the comments and modified our manuscript (attached) accordingly to the reviewer’s suggestion  The modifications appear in red color in the text.

the responses to the valuable comments are in the attached file

Reviewer 2 Report

This paper presents the control of a grid connected generator with a doubly salient exterior rotor excited by permanent magnets inserted in the stator yoke. Fuzzy logic is used in order to control the dq currents of the machine. The paper is generally well written, figure quality is good, and I believe a good amount of results is presented. As such, I congratulate the authors on their work. That being said, I have some comments and concerns about the manuscript, with the main one being the fact that only simulation results are presented. Please find my comments as follows:

1.      The contributions of the paper are not clear. Consider listing them at the end of the introduction section.

2.      Some uncommon English terms are used. Please go over the manuscript and current any instances of this. As an example, you use “calculus” in Figure 3, rather than calculation.

3.      In Figure 4 the Y axis is named as “quadratic current”, where it should actually be “quadrature current”.

4.      Provide comments on the robustness of the grid connected converter control algorithm. How does it behave in the face of parametric variations.

5.      Provide a legend in Figure 11(b) as one cannot identify the reference from the actual current.

6.      My main concern is that this paper only presents simulation results. For publication in journals of the field, experimental analysis and validation is often a must. I could not find anything on Actuators website regarding this, but in case this is allowed, I strongly encourage authors to pursue experimental results for future publications.

Author Response

The authors thank the reviewer for the valuable comments. We have considered all the comments and modified our manuscript (attached) accordingly to the reviewer’s suggestion. The modifications appear in red color in the text.

you will find the responses to the comments in the attached file  

Round 2

Reviewer 1 Report

Revision is satisfactory.
Please check again the FFT result in Figure 16-(a), expected THD is above 20%.

Author Response

Thank you for your comment and your positive analysis. Your comments were very helpful for the paper's improvement.

Reviewer 2 Report

Authors have replied to my comments in a satisfactory manner. Moreover, I believe the overall quality of the paper has been improved. As such, I believe the paper is now suitable for publication.

Author Response

(The authors gave the same response as above.)
